# Considerations of Control Groups: Comparing Active-Control with No Treatment for Examining the Effects of Brief Intervention

**DOI:** 10.3390/sports9110156

**Published:** 2021-11-19

**Authors:** Andrew M. Lane, Chris J. Beedie, Tracey J. Devonport, Andrew P. Friesen

**Affiliations:** 1Research Centre for Sport, Physical Activity (SPARC) School of Sport, Faculty of Education, Health and Wellbeing, Walsall Campus, University of Wolverhampton, Walsall WS1 3BD, UK; T.Devonport@wlv.ac.uk; 2School of Psychology, Canterbury Campus, University of Kent, Canterbury CT2 7NP, UK; C.Beedie-859@kent.ac.uk; 3Department of Kinesiology, Berks Campus, Pennsylvania State University, Berks, PA 19610, USA; axf716@psu.edu

**Keywords:** self-regulation, beliefs, intervention, performance, motivation, emotion

## Abstract

Background: A large-scale online study completed by this research team found that brief psychological interventions were associated with high-intensity pleasant emotions and predicted performance. The present study extends this work using data from participants (*n* = 3376) who completed all self-report data and engaged in a performance task but who did not engage with an intervention or control condition and therefore present as an opportunistic no-treatment group. Methods: 41,720 participants were selected from the process and outcome focus goals intervention groups, which were the successful interventions (*n* = 30,096), active-control (*n* = 3039), and no-treatment (*n* = 8585). Participants completed a competitive task four times: first as practice, second to establish a baseline, third following an opportunity to complete a brief psychological skills intervention, and lastly following an opportunity to repeat the intervention. Repeated measures MANOVA indicated that over four performance rounds, the intensity of positive emotions increased, performance improved, and the amount of effort participants exerted increased; however, these increases were significantly smaller in the no-treatment group. Conclusions: Findings suggest that not engaging in active training conditions had negative effects. We suggest that these findings have implications for the development and deployment of online interventions.

## 1. Introduction

The effectiveness of psychological skills such as imagery, goal-setting, and self-talk has been demonstrated in many areas of application [1], including sport [2], surgery [3], and computer gaming [4]. A recent large-scale study of 44,742 participants found support for the utility of following brief online active psychological skills training to aid emotion regulation and improve performance in a competitive task [5]. The aforementioned study [5] tested the effects of three psychological skills: (a) imagery, (b) self-talk, and (c) if–then planning, with each skill directed to one of four different foci: (a) outcome goal, (b) process goal, (c) instruction, or (d) arousal control, resulting in 12 different techniques. A 13th group labelled as a control group received a repetition of instructions on how to perform the task from Olympic gold-medallist Michael Johnson. The argument for labelling these participants as a control group was that they received no active training. They [5] compared the extent to which performance in the 12 intervention conditions improved over four rounds against the control group. The results illustrated the benefits of engaging in active psychological skills training, and the control group significantly improved also. Interestingly, the control group showed greater improvement in performance, felt more energetic, and exerted more mental effort than participants following instructional interventions.

A key aspect of this study [5] was the method used to produce the active control group which is used to form the case for the present study. In their study, participants were informed that they would learn about sport psychology and receive personalized feedback from Michael Johnson. Specifically, control group participants were informed, “You have played the game now. You have to find the numbers and finding them can be challenging. It’s a different grid but the challenges will be similar. Spend some time getting mentally ready. Give yourself about 90 seconds to prepare before you start the next round”. Although not receiving specific instructions, the control group received encouragement to perform again from former Olympian Michael Johnson, and encouragement is motivational [6].

A control group should seek to control the positive beliefs of using the intervention, a point that drives blind and double-blind placebo groups. The control condition should elicit some of the symptoms of the intervention but not those that are in the active treatment (e.g., decaffeinated coffee, a treatment that tastes like coffee, and so could have the active ingredient, but actually does not). In sport psychology interventions these typically involve active training, and as such it is difficult to have a traditional control group.

The present study extends this work [5] using previously unreported data from the same experiment. The investigators [5] found many participants engaged in all the performance tests but did not engage with the interventions. These unused data represent a novel condition and offers opportunistic no-treatment control data against the active control and active-training groups used in the previous study [5]. We hypothesized that the “no-treatment” group would perform significantly worse than the “active-training” and “active-control” groups reported previously [5].

## 2. Materials and Methods

### 2.1. Participants

Participants were 74,204 volunteers who provided informed consent and were recruited to the study via the British Broadcasting Corporation (BBC) Lab UK (*M*_age_ = 34.66 years, *SD* = 14.13). The project was advertised on national television and radio as an online experiment investigating performing under pressure. Participants originated from 103 different countries covering all continents. In the present study, we selected 41,720 (*M*_age_ = 34.34, SD = 13.93) participants from the process and outcome focus goals interventions, which were the successful interventions (*n* = 30,096; *M*_age_ = 34.64, SD = 14.07), active-control (*n* = 3039, *M*_age_ = 31.50, SD = 13.41), and no-treatment (*n* = 8585, *M*_age_ = 34.35, SD = 13.93).

### 2.2. Measures

The study uses the same measures reported previously [5] and so these are described only briefly here.

#### 2.2.1. Emotion

The items to measure, “Happy”, “Anxious”, “Dejected”, “Angry”, and “Excited”, were used from the same-named factors in the Sport Emotion Questionnaire (SEQ) [7] and two items “Fatigued” and “Energetic” were included to reflect arousal [8]. Each item was rated on a 7-point Likert scale (1 = *not at all* to 7 = *extremely)*. A single measure of emotion was used so that a high score was indicative of pleasant emotion. Alpha coefficients for emotion at each completion were: Baseline α = 0.72, Round 1 α = 0.70, Round 2 α = 0.68, and Round 3 α = 0.70.

#### 2.2.2. Concentration Game Task

A cognitive task was developed to allow the capture of a large dataset via an online method. The concentration grid task required participants to find and click on numbers in sequence from 1 to 36 as quickly as possible from a 6 × 6 grid. Numbers were presented in a randomised order within the grid, and as such participants had to concentrate and scan the grid to locate and click on the correct number. Participants completed a practice round, where participants performed alone and not against a competitor. Based on practice round results, an artificial computer opponent was introduced to create a sense of competition. The computer opponent was matched against the participant’s grid completion time from the practice round. The participant’s performance was measured by the number of seconds required to complete the grid.

#### 2.2.3. Mental Effort

The Rating Scale of Mental Effort [9] is a single item scale that was used to assess mental effort (0 = no effort to 150 = complete effort).

#### 2.2.4. Procedure

Data were collected online via the BBC Lab UK website. Participants completed informed consent forms before proceeding to the start of the online experiment. Videos guided participants through the completion of self-report scales and the concentration task. The online programme was narrated by Michael Johnson. Random allocation to experimental treatment groups was completed automatically by an online programme based on demographic data provided by participants. All participants completed the concentration game task before group allocation to provide a baseline measure of performance that could be used to assess whether the groups had pre-existing differences. Participants then rated their mental effort immediately following performance.

An opportunistic no-treatment group (*n* = 8595) emerged which consisted of participants who chose not to view the allocated intervention or encouragement video (i.e., active-control group). Instead, these participants immediately proceeded to a second completion of the concentration grid. Further, this was their decision. Therefore, considerations that arise when positive treatment is denied to a subsection of the sample in a randomised control design are not applicable. This no-treatment group is closer to a traditional control group. However, a key difference is that participants were not randomly allocated.

#### 2.2.5. Data Analysis

A repeated measures multivariate analysis of variance examining emotions, effort exerted and performance over the 4 rounds of practice, baseline, the implementation of the intervention, and finally, a repeat of the same intervention. The rationale for the data analysis strategy was to run as few tests as possible. With such a large sample size, it is easy to show significant results even though the size of the effect was low. In the present study, the focus is on significant interaction effects as they show that changes in data vary between groups.

## 3. Results

Repeated MANOVA results revealed a significant intervention effect (Wilks lambda _18,83522_ = 0.98, *p* < 0.0001, partial eta^2^ = 0.10), a main effect for changes over time (Wilks lambda _9,41769_ = 0.74, *p* < 0.0001, partial eta^2^ = 0.26) and a main effect for active-training, active-control, and no-treatment (Wilks lambda _18,83522_ = 0.98, *p* < 0.0001, partial eta^2^ = 0.11).

Univariate results indicated that emotions became significantly more positive (*F* _6,125304_ = 1328.56, *p* < 0.0001, partial eta^2^ = 0.03) following the completion of the intervention (see Figure 1). The pattern of significant differences showed that the active-training group benefited the most, followed by the active-control group, with the least benefits being found for the no-treatment group. Weaker significant interaction effects were found for effort invested in performance over rounds of competition (*F* _6,125304_ = 12.46, *p* < 0.0001, partial eta^2^ = 0.01, see Figure 2) and for improvements in performance (*F* _6,125304_ = 6.42 *p* < 0.0001, partial eta^2^ = 0.001, see Figure 3).

Results show that there were main effects for time (*F* _3,120856_ = 1328.56, *p* = <0.001, partial eta^2^ = 0.007) with emotions became significantly more positive (*F*
_3,1328.56_ = 2132.93, *p* < 0.0001, partial eta^2^ = 0.49), performance improving (*F* _3,120856_ = 1249.97, *p* = <0.001, partial eta^2^ = 0.007) and effort invested (*F* _3,120856_ = 4836.33, *p* = <0.001, partial eta^2^ = 0.104).

## 4. Discussion

The present study examined the effects of brief online interventions in comparison to a no-treatment group [5]. The large sample of data in the active-training and active-control groups offer a good opportunity to conduct a comprehensive examination. A total of 8585 participants did not complete the interventions but engaged fully in other parts of the experiment and therefore represent what typically looks like a non-treatment group. The key difference between these participants and a traditional control group was they were not randomly allocated. Participants in the no-treatment group showed performance improvement from baseline, increased intensity of positive emotions, and increased effort. This result is interpreted as suggesting participants in the no-treatment group were motivated to improve performance, and therefore resemble participants who sign up with a desire to improve performance. However, the rate of change between rounds was slower for the no-treatment group than the active-training and active-control groups, suggesting the active part of either control or training was influential. Compliance with participation protocols is a key factor when examining the effectiveness of interventions. In research, participants who do not comply with protocols is an issue. In real world settings, poor participant compliance minimizes the effectiveness of treatments ranging from COVID-19 vaccines to physical and mental health interventions.

Results demonstrated that the no-treatment group performed significantly worse, made less progress, and reported less optimal psychological states than the active-control and active-treatment groups. These results are not entirely unexpected but exploring possible reasons why they occurred and learning from using online interventions where naturally occurring no-treatment groups could emerge could have useful implications for future work. Positive benefits from participants receiving treatment could be explained by enhanced beliefs that the treatment would work, an effect normally described as a placebo effect. Controlling beliefs is typically achieved by using a blind placebo approach. However, this is not possible where an intervention requires the person to act consciously on information provided. A blind placebo arguably works much better in studies such as caffeine where people still participate in the treatment, believing they are taking the caffeine, but the active ingredients are removed. In such research, great care is made to make a placebo look like an authentic treatment. In a sport psychology intervention where a practitioner teaches the use of psychological skills, there is a requirement for the participant to be active in the process. An active-control group [5] is one in which basic instructions are repeated and so attempts to control for belief effects. The present study which used a no-treatment group offers the opportunity to compare the effects of active treatments against no-treatment. A no-treatment group resembles what happens in real life when people wish to pick up a skill and learn by trial and error and without specific guidance.

The active-control group benefited from participation in the study more than the no-treatment group. Receiving a message from an inspirational figure such as Michael Johnson and expecting personalised feedback can be argued to provide encouragement, which is motivating [6]. Whilst encouragement is a simple technique to use as an intervention, it is possible that the effectiveness of it in this context derives from it being delivered by a highly influential figure in sport. This raises the issue of the relative influence of the person who delivers the intervention as an active ingredient. Models of social influence from social psychology have highlighted the importance of the perceived status of the influencer [10]; however, this issue is under-examined within the context of conducting psychology research. We suggest future research compares the effectiveness of encouragement when the same message is given by different people, with a hypothesis that the more credible the persuader, the more influential it would be.

The current study can also inform future research that uses online methods to investigate psychology interventions. Online data collection that allows people to volitionally skip through the intervention creates naturally occurring control conditions where participants do not expect the intervention to work. In the present study, the no-treatment group was an opportunistic group that emerged once data had been collected and so differed from a traditional control group for whom the condition was randomised.

The present study offers a valuable contribution to knowledge in this area. Results show the value of online research which offers scalability and via data capture processes can facilitate the examination of points of engagement with the task and intervention being delivered. However, we recognise at least two limitations. The first limitation is that we did not obtain feedback from participants that measured any learning effects having completed the intervention. That is, we did not check to see if the knowledge gained was internalised before starting the task. The second limitation is that participants were not randomly allocated into the no-treatment group. We should not see engagement and disengagement as dichotomous concepts and appreciate that the intensity with which people engage with the intervention will vary. A limitation with online studies is that the conditions in which a person learns an intervention and takes a test has many unknown features. We suggest that future research focus on the learning process in terms of what intervention content is retained.

In conclusion, the BBC Lab UK data show the benefits of capturing all keyboard data. The present study used data that were not used previously [5]. On initial analysis these data were seen as incomplete; however, this shows the benefits of reflecting on what insights such data might provide. We encourage researchers to focus on using entire datasets to interrogate the issue of compliance in completing interventions, and to investigate why non-compliance occurs.

## Figures and Tables

**Figure 1 sports-09-00156-f001:**
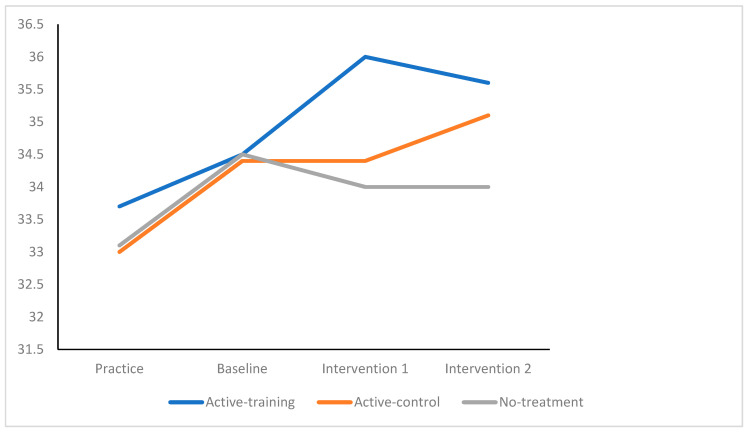
Emotion by Competition Rounds, by Active-Training, Active-Control, and No-Treatment.

**Figure 2 sports-09-00156-f002:**
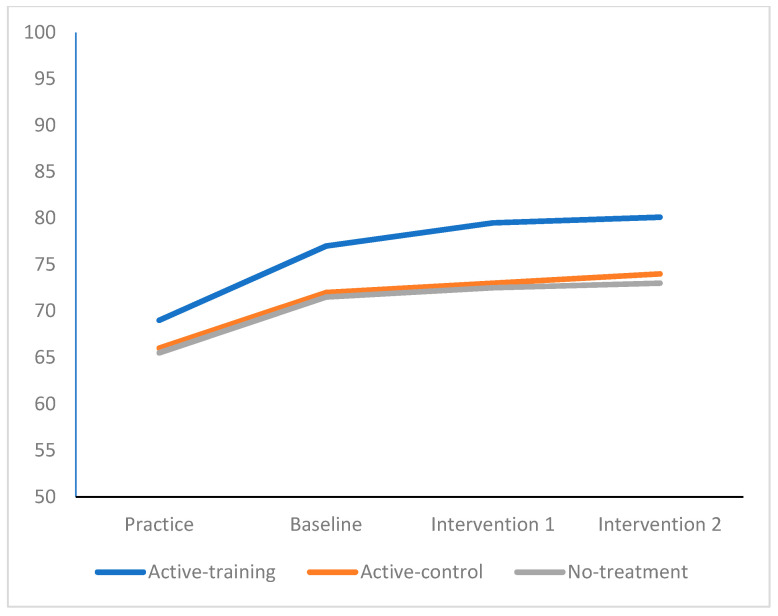
Effort by Competition Round, by Active-Training, Active-control, and No-Treatment.

**Figure 3 sports-09-00156-f003:**
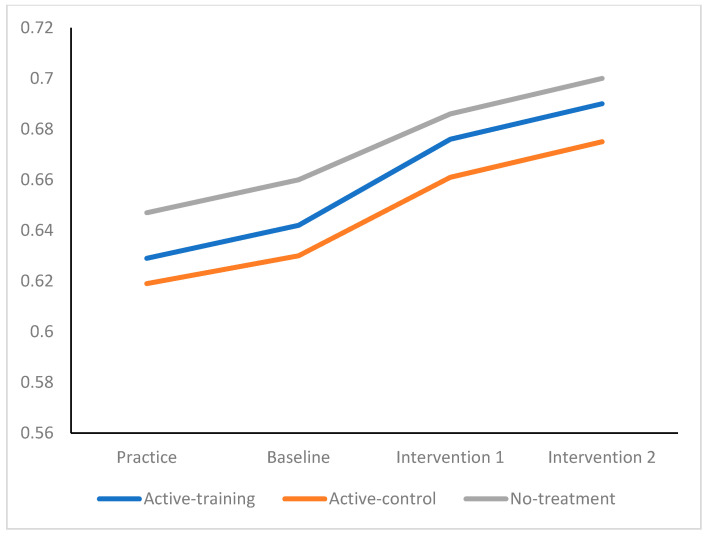
Performance by Competition Round, by Active-Training, Active-Control, and No-Treatment.

## Data Availability

The data is not yet publicly available.

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
