# Peer review of "Considerations of Control Groups: Comparing Active-Control with No Treatment for Examining the Effects of Brief Intervention"

_sports, 2021, doi:10.3390/sports9110156_

Round 1
Reviewer 1 Report
I would like to thank the editorial board for the opportunity to review the current manuscript that assessed “Considerations of Control Groups: Comparing Active-Control with No treatment for examining the effects of brief intervention”. The current investigation extends the work from Lane, Totterdell, et al. (2016). This study provides interesting findings which may have an influence on the design of future intervention studies. I only have a minor comment to make, which I have outlined below.
Comments
General
Figure 1-3 – the quality of resolution of the figure should be improved. Please check the journal’s specifications for the resolution required. Please include the X and Y axis within each figure. I suggest removing the horizontal lines and choose a line graph with points to highlight practice, baseline, and each intervention.
Author contributions should be completed
Author Response
Many thanks for your positive comments. We have re-drawn the graph and the resolutions should be better. We justified the data analysis and sought to make the article read better. We used track changes and so it is easy to see the changes.

Reviewer 2 Report
I commend the authors on their decision to explore this unused data in a different and insightful manner. This novel opportunity is well presented and provides great detail that may not have been available previously. The writing throughout is clear and succinct. There are a possibly a couple of places where the text could be slightly improved, but a re-read by the authors will be sufficient to identify these. Overall I cannot fault the work presented.
Author Response
Many thanks for your positive comments. We have been through the article and hopefully tidied up the quality of writing.
